# Virological investigation and comparative genomic analysis of elephant endotheliotropic herpesvirus 1B infection in an Australian captive herd of Asian elephants (*Elephas maximus*)

Jack W. Wheelahan[ID][1]*, Paola K. Vaz[ID][1], Alistair R. Legione[ID][1], Carol A. Hartley[1], Natalie L. Rourke[2], Michael Lynch[3,4], Bonnie McMeekin[3], Elizabeth C. Dobson[5], Joanne M. Devlin[1]

**1** Asia-Pacific Centre for Animal Health, Melbourne Veterinary School, Faculty of Science, The University of Melbourne, Parkville, Victoria, Australia, **2** Werribee Open Range Zoo Veterinary Department, Werribee, Victoria, Australia, **3** Melbourne Zoo Veterinary Department, Parkville, Victoria, Australia, **4** Melbourne Veterinary School, Faculty of Science, The University of Melbourne, Werribee, Victoria, Australia, **5** Veterinary Anatomic Pathology, Melbourne Veterinary School, Faculty of Science, The University of Melbourne, Werribee, Victoria, Australia

* jack.wheelahan@unimelb.edu.au

## Abstract

Elephant endotheliotropic herpesviruses (EEHV) pose a significant threat to the conservation of Asian elephants (*Elephas maximus*) worldwide, with a high mortality rate in young elephants. However, several components of EEHV virology remain underexplored, particularly for EEHV1B. This study describes a fatal case of EEHV1B infection in a nine-year-old Asian elephant from an *ex situ* conservation herd, examining herd viral dynamics, tissue viral loads and comparative genomics. This elephant succumbed to haemorrhagic disease within three days of developing clinical signs, despite therapeutic intervention. Quantitative PCR (qPCR) was performed on serial trunk washes and whole-blood surveillance samples collected before and after the clinical event, as well as on post-mortem tissues preserved in different storage media (DNA/RNA Shield, RNALater, and viral transport medium). Metagenomic next-generation sequencing of infected tissues was performed to characterise the complete viral genome, analyse variation from other published EEHV genomes and assess for evidence of viral recombination between EEHV subspecies. The affected elephant demonstrated a marked viraemia at onset of clinical disease, with viral load peaking at $5.47 \times 10^6$ viral genome equivalents per mL of blood, one day after the onset of clinical signs. Samples stored in viral transport medium yielded the greatest viral and host DNA recovery by qPCR, although tissues stored at −80 °C without media were still suitable for molecular detection. Whole genome sequencing demonstrated 96.0% pairwise nucleotide identity between the assembled genome (EEHV1B_AUP_01_2023, GenBank accession: PX651398) and the previously reported EEHV1B sequence (KC462164), and a maximum of 90.9% identity to

**Data availability statement:** All raw sequencing data generated in this study have been deposited in the Sequence Read Archive (SRA) under BioProject ID: PRJNA1366724. The full genome sequence EEHV1B_AUP_01_2023 has been submitted to Genbank and is available under the accession number: PX651398. All other data are in the manuscript and/or supporting information files.

**Funding:** This study was funded by generous donation from the Cybec Foundation to the University of Melbourne (J.D., J.W.) (https://www.cybec.org/), by the Australian Research Training Program Scholarship (Stipend) (J.W.), Australian Government (doi.org/10.82133/C42F-K220), and by the Ernie Stewart Memorial Scholarship (J.W.), University of Melbourne (https://scholarships.unimelb.edu.au/awards/ernie-stewart-memorial-scholarship). Funding sources were not involved in study design, collection, analysis or interpretation of data, nor writing of the report.

**Competing interests:** The authors have declared that no competing interests exist.

published EEHV1A genomes, with evidence of recombination between the viral subspecies at several genomic regions. Viral recombination between EEHV subspecies may have significant implications for the pathogenesis of EEHV disease, the reliability of molecular diagnostics and the efficacy of vaccinations and anti-viral therapy.

## Introduction

Elephant endotheliotropic herpesvirus (EEHV) infection can result in haemorrhagic disease (EEHV-HD) and death in Asian elephants (*Elephas maximus*) [1,2]. Clinical cases of EEHV-HD have been reported primarily in Western zoos, but also within wild and semi-captive elephant populations of endemic countries [3–5]. Approximately 20% of Asian elephants in captivity develop EEHV-HD [6]. Diagnosis is made by the presence of clinical signs and/or haematological abnormalities coinciding with the presence of EEHV viraemia detected using molecular methods [2]. The mortality rate of individuals developing EEHV-HD is 68–85% [4,7]. There have been at least 58 fatal EEHV-HD cases in Asian elephants from European and North American zoos [8,9], and over 100 confirmed fatal cases from endemic range countries (both in captive and wild animals) [9–12]. Young elephants are disproportionately affected by this disease, with the key risk period being between two and eight years of age [6]. This risk period coincides with waning of protective maternal antibodies and primary virus exposure [13,14]. Delayed viral exposure due to small herd sizes in captivity is hypothesised to be the reason for an increased risk of disease in captivity [13]. Fatal disease therefore poses a significant risk to both *ex situ* conservation programs with declining captive populations, and to *in situ* conservation efforts where elephant populations are becoming increasingly fragmented [12,15,16].

These viruses belong to the subfamily *Betaherpesvirinae*, genus *Proboscivirus* [1,17]. Several EEHV species have been described (EEHV1–7), with further subspecies designation for some EEHV species (e.g., EEHV1A and EEHV1B) [18–20]. In Asian elephants, EEHV-HD morbidity and mortality has been associated primarily with EEHV1A infection, but also observed in cases of EEHV1B, EEHV4 and EEHV5 infection [4,8]. Each of the EEHVs may be detected by quantitative polymerase chain reaction (qPCR) in circulating blood of infected individuals, and in trunk washes of elephants shedding the virus [21–23]. Further differentiation of EEHV1 subspecies can be performed by subsequent qPCR testing with subspecies specific gene targets [23]. Detection of EEHV virus in circulating blood can precede the onset of clinical signs by up to one week [24]. Early detection of viraemia and timely therapeutic intervention is associated with cases surviving EEHV-HD [4,10]. Regular surveillance by qPCR for viraemia can therefore facilitate rapid diagnostic and therapeutic intervention when indicated. However, intermittent viraemia and viral shedding can occur for months to years after an individual experiences a primary infection [25,26]. Understanding which individuals are vulnerable to primary infection, and monitoring the viral dynamics of the herd, are therefore key to successful management of EEHV-HD risk.

Viral whole genome sequencing has provided a means of investigating EEHVs, particularly in the absence of successful viral cultivation [6,27,28]. Several whole and partial genome sequences have now been published for each of the EEHV species [17,20,27–30], although only one previous EEHV1B whole genome sequence has been published [18]. This EEHV1B reference genome was from necropsy tissue of a two-year-old elephant that died at Whipsnade Zoo, United Kingdom in 2006. In addition to this single whole genome sequence, partial genome sequences have been published for six additional EEHV1B cases (3.6 kb – 54.7 kb, median 32.3 kb) [17,28]. In contrast, there are at least eight published EEHV1A whole genome sequences and partial genome sequences from more than 25 additional EEHV1A cases (1.1–179.8 kb, median 11 kb).

The EEHV genomes are structured as two terminal repeats flanking a single unique long region, which is divided into segments L1, L2, L3, C1, C2, R1 and R2, where the core herpesvirus genes are contained within regions C1 and C2 [18,29,31]. EEHV1A and EEHV1B are two subspecies that share a relatively high percentage (~90%) nucleotide identity across the whole genome but differ primarily at three chimeric domain regions (CD-I, CD-II, CD-III) and at several hypervariable regions including the R2 segment [17]. These chimeric domain regions encompass part or all of 12 genes across 15 kb of the genome, and are hypothesised to have arisen by an ancient recombination event between two highly divergent genomes [17]. Importantly, these chimeric domain regions include key envelope glycoproteins that likely facilitate cell entry including glycoprotein B (gB), glycoprotein H (gH), glycoprotein L (gL) and three adjacent spliced glycoproteins ORF-O, ORF-P and ORF-Q [27]. Infection with either EEHV1 subspecies does not provide immunological protection against subsequent infection with the other subspecies (EEHV1A infection following EEHV1B or vice versa) [32]. Examining genomic differences between EEHV1 subspecies therefore may unveil key viral antigens targeted by the host humoral and adaptive immune systems. Such differences may also play a role in the difference in the pathogenicity, morbidity and mortality associated with each subspecies, where EEHV1A is a much more common cause of fatal disease than EEHV1B [6].

Despite EEHVs being widely reported, no cases of EEHV-HD arising within Australian elephant herds have previously been described in the scientific literature although a single fatal case of EEHV-HD occurred in 2018 at Taronga Zoo, Sydney, resulting from an EEHV-4 infection [33]. Australia currently hosts 27 Asian elephants in captivity across five locations: Werribee Open Range Zoo (n = 9), Taronga Western Plains Zoo (n = 7), Monarto Zoo (n = 5), Australia Zoo (n = 4) and Sydney Zoo (n = 2). Five of these elephants are considered at high risk for EEHV-HD based on their age (between one and eight years of age) [34]. Host genetic factors and viral prevalence and genomics may vary between sites and could contribute to disease susceptibility, as is seen in other herpesviruses [35]. Documentation of EEHV-HD clinical cases from a new geographical region may therefore elicit unique clinical, pathological or genomic findings. Similarly, documenting cases with unusual features (in this example, in an animal older than the recognised high-risk age group) also contributes new information.

This report documents EEHV1B infection in a captive herd of Asian elephants in Australia, associated with a fatal case of EEHV-HD in a nine-year-old male elephant. Viral shedding dynamics within the herd are described through molecular detection of viral DNA in trunk wash samples collected over time. An investigation into virus tissue tropism in the affected animal is reported using molecular detection of viral DNA in post-mortem tissue samples collected into different storage media. Next-generation sequencing techniques were applied to tissues with high viral load to obtain viral DNA sequence information for comparative genomic and phylogenetic analysis.

## Materials and methods

### Herd information and individual clinical data

The elephant herd described in this report is held in captivity by Zoos Victoria, and consisted of five adult females (13 years to 50 years of age), one adult male (14 years of age), one sub-adult male (nine years of age) and three juvenile calves (less than one year of age) at the time of this clinical event (individual demographic data in S1 Table). The clinical

case described here occurred in the nine-year-old male (identified as Elephant 7 in this case), who had previously demonstrated diagnostic positives for EEHV1 (undetermined subspecies) and EEHV1A in blood and trunk wash samples. Clinical information from this case, including onset and progression of clinical signs, haematology and biochemistry analyses, and therapeutic intervention will be reported separately. A brief clinical synopsis for context is provided in this report. A full post-mortem examination, including histopathology, was performed by a specialist veterinary pathologist and is summarised here.

## Sample collection

Trunk wash samples from herd members were collected fortnightly for assessment of EEHV shedding within the herd as previously described by Stanton, Zong [25]. Briefly, 50 mL of sterile saline (0.9% w/v NaCl) were poured into the nares of each elephant, the proboscis was elevated for approximately 30 seconds, and elephants then expelled the nasal contents into a fresh plastic freezer bag on command. Samples were transferred from these bags into sterile 50 mL screw cap conical centrifuge tubes, centrifuged at 1500 x $g$ for 5 minutes at room temperature; the supernatant was discarded, and cell pellets were stored at −20 °C until further processing for DNA extraction. All available samples were collected three months prior to, and 12 months after the death of the affected elephant and these were tested for viral DNA as part of routine veterinary care, described further below.

Whole blood samples were collected from elephants in the herd weekly until eight years of age, fortnightly until nine years of age and monthly from nine years of age, or at any sign of illness. Blood samples at the time of illness were submitted to the closest laboratory with EEHV diagnostic capabilities: Veterinary Pathology Diagnostic Services at the University of Sydney, Australia. DNA from archived whole blood samples from this time-point was subsequently extracted and re-tested after the death of the elephant to measure changes in viraemia over time.

A full post-mortem examination was performed approximately 48 hours after death and tissue samples from a range of different organs (approximately 1 g of each tissue) were collected for virological analysis, including organs for which EEHV1 has known tropism. These samples were stored in each of DNA/RNA Shield™ (Zymo Research, Irvine, CA, USA), *RNALater*® Stabilization solution (Thermo Fisher Scientific, Waltham, MA, USA), a commercial viral transport medium UTM® Universal Transport Medium™ (Copan, Murrieta, CA, USA), or in plain tubes with no storage media. Approximately 3 mL of preservative solution were added to each tissue sample, and all tissue samples were stored at −80 °C until processing for DNA extraction.

Sample collection and testing within this study was approved by the Zoos Victoria Animal Ethics Committee (project reference number ZV23019).

## DNA extraction

DNA extraction of trunk wash and whole blood samples was performed using the Extracta Plus DNA kit (QuantaBio, Massachusetts, USA). Trunk wash samples were processed as per the manufacturer's tissue homogenate extraction protocol with 8–12-hour overnight incubation with tissue lysis buffer and proteinase K, whereas whole blood samples were processed as per the manufacturer's whole blood extraction protocol. DNA from tissue samples was extracted with the Wizard® SV Genomic DNA Purification System (Promega, Wisconsin, USA) using approximately 20 mg of stored tissue sample, as per manufacturer's instructions.

## Virus quantification by qPCR

Extracted DNA samples were tested by hydrolysis probe qPCR for the presence of EEHV1, 4 and 5 [21,25], as well as a host reference gene TNF-$\alpha$ (Table 1). As the EEHV1 qPCR does not differentiate EEHV1A from EEHV1B, subtyping EEHV1A and 1B qPCR assays were performed if the initial EEHV1 assay returned a positive result [23]. Different qPCR platforms were used to test tissue samples compared with the whole blood and trunk wash samples due to differences

**Table 1. Molecular detection assay targets for each EEHV (sub)species and elephant TNF-alpha.**

| Assay | Target region | Gene locus | Sequence (5' – 3') | Reference publication |
|---|---|---|---|---|
| Elephant host DNA | Tumour-necrosis factor- alpha (TNF-α) | NA | **CCCATCTACCTGGGAGGAGTCT**TCCAGCTAGAGAAGGGTGATCGACTCAGCGCT-GA**GATCAATCTGCCTGACTATCTCGA** | Stanton, Zong [24] |
| EEHV1 | Major DNA-binding region | U41 | **CGATGATACCCGATCCCTAGTC**CGACAGCACACCGCAAAACCAAAAAATCTTA-AATTCAATATGGAT**CATCTAAGCTTCGGCGCC** | Stanton, Zong [25] |
| EEHV1A | Myristylated Tegument Protein (MyrTeg) | U71 | **TGAGAACGAATACTTGGAGCATTG**CTGCAAATCGTCCCATCGCAGCCCTTCT-CATCGTGGAT**GTACGATGTCACTCCAGGTGTAAC** | Pursell, Tan [23] |
| EEHV1B | Glycoprotein H (gH) | U48 | **TACGCCAGGTGTAACATCCTATATACA**CAACGAAAATGTTGATACGGCGACGTGC-GCTGTGGCTGACAATCCCCATTTTAATCA**TCATGACGTCTCATAAAATAATGTGC** | Pursell, Tan [23] |
| EEHV4 | Glycoprotein B (gB) | U39 | **AGCTATCGGAAGACGATGTCA**ACGCACCGCTCGTTACCGAAGAATGTGT**CAT-CACCACAGAAGTAATCACCG** | Pursell, Tan [23] |
| EEHV5 | DNA Polymerase | U38 | **CCTGGTTGGCGGAAAGAA**AAGCCGTGAGAGAAAAGTTAA**AACAGTGTAGTGA-CCCTTTGATGC** | Stanton, Nofs [21] |

Sequences are provided 5' to 3' based on annotated gene direction. Primer and probe sequences are bolded and underlined respectively.

in available instruments across laboratories where the samples were tested. Extracted DNA from tissues samples were tested using an AriaMX qPCR machine (Agilent Technologies, Santa Clara, USA). Extracted DNA from trunk wash and whole blood samples were tested using a Mic qPCR machine (Biomolecular Systems, Gold Coast, Australia). Master mix constituents for each assay included 0.9 µM of each forward and reverse primer, 0.25 µM of FAM-labelled fluorescent probe, 10 µL of PerfeCTa® qPCR ToughMix® UNG (QuantaBio, Massachusetts, USA), 5 µL of DNA template and the remaining volume to 20 µL made up with sterile water. Negative control reactions contained no DNA template and negative DNA extraction controls were included in all qPCRs. Thermal cycling was performed as previously described [25], with holds at 50 °C for 2 minutes and 95 °C for 10 minutes, followed by 40 cycles of 95 °C for 15 seconds and 60 °C for 60 seconds. Amplification curves were compared to standard curves generated by means of qPCR on serial 10-fold dilutions of a known quantity of synthetic DNA constructs of the target region (see Table 1), tested in triplicate. Standard curves were generated from $2 \times 10^8$ to $2 \times 10^2$ copies/µL for tissue samples, and from $2 \times 10^6$ to $2 \times 10^0$ copies/µL for trunk wash and whole blood samples. Cut-off for positive results was determined by the lowest concentration of the standard curve to test positive for all three replicates. Quantification of EEHV1 in positive samples was determined using the EEHV1 qPCR rather than the specific EEHV1A/B subtype qPCR assays. For tissue samples, the quantity of elephant cells was calculated by dividing the determined copies of TNF-α by two, to account for the presence of two gene copies per diploid cell in the elephant genome. Calculated viral genome quantities were then normalised to host cell number (viral genome copies/cell) by dividing the VGE/µL by the cells/µL for each sample.

## Sequencing and genome assembly

Extracted DNA from heart and intestinal tissue were processed using Genomic DNA Clean and Concentrator™ –10 (Zymo Research, Irvine, CA, USA) following the manufacturer's instructions. Samples were submitted for Novaseq (Illumina) metagenomic next-generation sequencing to the Australian Genome Research Facility, Parkville, Victoria, Australia. Reads were taxonomically classified using the tool Centrifuge tool v1.0.4 [36] through read alignment to a curated NCBI nucleotide database [37]. Data was processed using Geneious Prime® 2025.0.3 (https://www.geneious.com) [38]. Short reads for mapping to reference were generated from metagenomic data by trimming to remove sequence adapters and low-quality reads (Phred score < 30) using the Geneious plug-in BBDuk Trimmer v1.0 [39]. The quality of paired end

sequencing reads was assessed using FastQC v0.12.0 [40]. Given the abundance of viral DNA in heart tissue, the full viral genome sequence was generated with this tissue. For *de novo* assembly, raw reads were deduplicated and trimmed to remove low quality bases using fastp v23.2 [41]. Trimmed reads were assembled into contigs using MEGAHIT v1.2.9 [42].

The full genome sequence of EEHV1B_AUP_01_2023 (herein referred to as AUP_01) was determined using a modified approach of that described by Sijmons, Thys [43] used for the assembly of HCMV genomes. The approach involved generating a preliminary consensus sequence from short reads mapped to the published EEHV1B reference genome: Emelia (GenBank accession: KC462164) [18], mapping *de novo* assembled contigs to this preliminary consensus to develop a hybrid scaffold; and then iteratively mapping short reads to generated consensus sequences until no additional reads were mapped. Short reads were mapped using minimap2 v2.24, allowing a maximum of three secondary alignments [44]. Contigs were mapped to the rearranged EEHV1B Emelia reference sequence using minimap2 with "Long assembly 20% divergence" input data setting available in Geneious. Iterative consensus sequences were generated from reads mapped using the in-built Geneious "Generate consensus" function, with a 0% majority threshold ignoring gaps. This approach was compared to iterative mapping of short reads to EEHV1B Emelia reference and subsequent consensus sequences without use of *de novo* assembled contigs using both minimap2 and an in-built Geneious mapper on 'Medium-Low Sensitivity' allowing for detection of structural variants, insertions and deletions of any size, as well as to mapping of *de novo* generated contigs only. Short reads from intestinal tissue were subsequently mapped to AUP_01 as the reference to compare the viral genome sequence and coverage detected in different tissues.

The published EEHV1B Emelia reference by Wilkie, Davison [18] is in opposite arrangement to other peer-reviewed EEHV1 whole genomes that adopt the convention defined by Ling, Reid [29] and followed by Richman, Zong [17]. The arrangement defined by Ling, Reid [29] more closely matches the genomic arrangement for other mammalian herpesvirus subfamilies [31]. Therefore, AUP_01 and EEHV1B Emelia reference whole genome sequence were aligned to the EEHV1A reference sequence (Kimba NAP23, GenBank accession: KC618527) using the Mauve alignment tool [45] and rearranged to match the EEHV1A arrangement. Genome annotation was performed using the EEHV1B Emelia genome (n = 103/119), and several EEHV1A sequences (n = 16/119) (MN366290, MN366293, OR543011, MH287519, MH287533) as references, with annotation nomenclature following the naming convention implemented by Ling, Reid [29] (see S2 Table for disambiguation). The complete genome was uploaded to GenBank (Accession: PX651398).

## Phylogenetic and genome variant analyses

Single nucleotide polymorphisms (SNPs) and multi-nucleotide variants (MNVs) between AUP_01 and the EEHV1B Emelia reference, as well as between AUP_01 and viral genome assembled from intestinal tissue, were found by using the 'Find Variations/SNPs' tool in Geneious on a pairwise alignment of the two sequences, allowing merging of adjacent mutations. Polymorphism annotations were then imported into R [46] for further analysis with the 'summarytools' package [47] and visualisation with the 'ggplot2' package [48]. Nucleotide polymorphism data were analysed by polymorphism type, protein effect, and SNP count per CDS region. Non-synonymous substitutions resulting in protein product changes were defined as any polymorphism resulting in amino acid substitution, insertion, deletion, frame shift, extension, loss of start codon or premature truncation. The ratio of non-synonymous (dN) to synonymous (dS) substitutions was determined by creating a codon-based alignment of coding sequences using PAL2NAL [49], inputting this alignment into KaKs calculator v2.0 [50], and estimated using the GY-HKY method [51]. Coding regions were included for dN/dS analysis if the sequences shared CDS annotations at the same site, if CDS regions were able to be codon aligned, and if the number of SNPs was ≥ 2.

Whole genome alignments between AUP_01 and EEHV1 genome sequences (EEHV1A: NAP23 Kimba [KC618527], EP22 Raman [KC462165], IP006 Nirangen [MW015025], IP043 Chellama Vandular [MN366291], IP091 Thirunelli1 [MN366290], IP143 Kozhikode1 [MN366294], IP164 Muthanga2 [MN366293], and Kanja (MN067515); EEHV1B: EP18 Emelia) were created using Multiple Alignment with Fast Fourier Transformation (MAFFT) v7.490 within Geneious [52]

using the default settings. Separate alignments of concatenated genes were also created using MAFFT for regions of interest including highly conserved genes identified by ICTV as best suited to use for phylogenetic analysis [53]. A maximum likelihood phylogenetic tree comparing reference genomes of each EEHV species sourced from NCBI Genbank (see S3 Table), to AUP_01 was constructed with PHYML v2.2.4 using GTR substitution model with 100 bootstrap replications [54]. This phylogenetic tree was constructed using a concatenation of conserved genes encoding DNA polymerase (U38), glycoprotein B (U39), major capsid protein (U57), terminase (U60), helicase (U77) and uracil DNA-glycosylase (U81) as recommended by Gatherer, Depledge [53].

### Recombination analysis of the EEHV1B assembled genome

Alignments of AUP_01 to EEHV1A and EEHV1B sequences were prepared using MAFFT and assessed for evidence of recombination using the phylogenetic network analyses program SplitsTree v6.4.13 [55]. This was performed on whole genome sequence alignments, on partial genome alignments of C1 and C2 regions, and on more divergent regions including the R2 region and focal regions of the L1 (E1 – E7) and L2 (E21 – E25) regions. Neighbor-net split networks were constructed using uncorrected $p$-distances and excluding gap sites using the 'ActiveSet' inference algorithm. Evidence for recombination was assessed in SplitsTree by performing pairwise homoplasy inference (PHI) statistical tests [56]. Sequence similarity plots between AUP_01 and EEHV1A and 1B reference genomes were performed using SimPlot++ v1.3 [57]. For whole genome analysis, the window size was set to 1000 nucleotides, and step size to 100 bp, whereas for partial genome analysis in divergent regions, the window size was set to 400 nucleotides, and step size to 40 bp.

## Results

### Severe viraemia at the onset of mild clinical signs

This fatal case of EEHV1B-HD occurred in a male elephant that was nine years and eight months old. Three weeks prior to the onset of illness, a routine whole blood sample was collected from the affected elephant and it tested negative for all EEHVs by qPCR. At first onset of illness (day 0) the elephant demonstrated mild lethargy, and blood was collected for haematology, biochemistry analysis and EEHV qPCR. This sample was submitted to the Veterinary Pathology Diagnostic Services at the University of Sydney, New South Wales, Australia, and detected an EEHV1 viraemia with an estimated viral load greater than $1 \times 10^5$ VGE/mL.

Clinical status declined over the following two days despite intensive therapeutic intervention according to the current EEHV advisory group treatment protocols [58]. Thrombocytopaenia and monocytopaenia, in addition to a rising viraemia (estimated at greater than $1 \times 10^6$ VGE/mL) were detected on day 2 of clinical disease, with oedema of the neck and face observed that evening. Subsequent testing of archived blood samples in a University of Melbourne laboratory from each day of treatment showed a viraemia of $4.21 \times 10^6$ VGE/mL on day 0, $4.77 \times 10^6$ VGE/mL on day 1 and $5.47 \times 10^6$ VGE/mL on day 2. The elephant succumbed to fatal EEHV-HD in the early morning of day 3.

### EEHV1B shedding increased in the entire herd after fatal case of EEHV-HD

Detection of viral DNA in herd trunk wash samples by qPCR showed EEHV1B in 20.7% of samples (n = 12/58) from seven animals prior to the onset of clinical disease in the affected animal (Elephant 7), compared with 44.4% of samples (n = 16/36) in the three months following the clinical case, and 23.4% (n = 26/112) in the subsequent nine months (Fig 1) (Fisher's exact test, $p = 0.03$). A spike in EEHV1B shedding (27,320 viral genome copies per reaction) in one elephant (Elephant 2) was detected seven weeks prior to the onset of clinical disease in Elephant 7. This elephant demonstrated high levels of shedding of EEHV1B across three samples every fortnight up until seven weeks prior to the clinical case. Trunk wash samples from this elephant however were negative for EEHV1B in the two samples collected fortnightly in the month preceding onset of clinical disease. Much lower levels of EEHV1B shedding (3.0–8.5

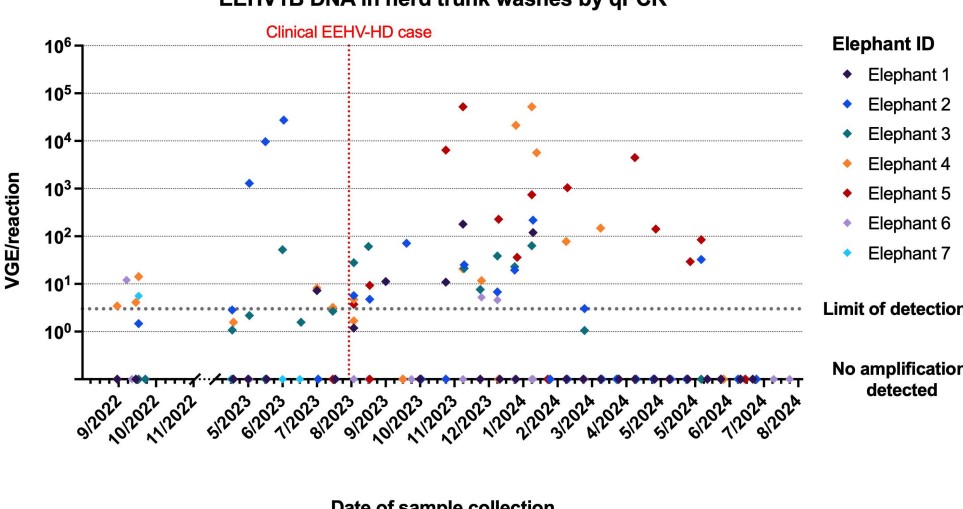

**Fig 1. EEHV1B viral genome equivalents (VGE) detected in trunk wash samples by qPCR [25] from elephants of the herd over time.** Different elephants are identified by diamond colour, and a fatal EEHV-HD case (Elephant 7) is outlined by the red dotted line. Viral quantities were determined by EEHV1 qPCR, and all EEHV1 positives were subsequently subtyped to EEHV1B. Sampling of trunk washes was not performed between November 2022 and May 2023 which is represented by the x-axis break.

VGE/reaction) were detected in two other elephants (Elephants 3 and 4) two weeks prior to the onset of disease, whereas the other four elephants (Elephants 1, 2, 5 and 6) tested negative for EEHVs at this time-point. In the weeks following the death of Elephant 7, trunk washes from all members of the herd tested positive to EEHV1B and remained positive in some individuals for up to 10 months. This was observed despite no development of clinical signs consistent with EEHV-HD, no haematological abnormalities, and only a single low-level viraemia detection (2,400 VGE/mL of blood) in one individual (Elephant 5) four months after the clinical case. No EEHV1A was detected within the study period. The detection of EEHV4 in trunk washes was marginally increased in the three months following disease with 8.3% of samples positive (n = 3/36) compared to 0% (n = 0/58) prior and 0% (n = 0/112) in the subsequent nine months (Fisher's exact test, $p = 0.005$). In contrast, shedding of EEHV5 was marginally higher prior to the clinical case with 17.2% (n = 10/58) compared to 5.6% (n = 2/36) in the three months and 2.7% (n = 3/112) in the subsequent nine months after the case (Fisher's exact test, $p = 0.003$).

## Pathology findings included marked lymphadenomegaly and lymphoid depletion

Gross pathology findings described were consistent with previous descriptions of fatal cases of EEHV-HD [59]. Key findings included lingual cyanosis, coalescing haemorrhage of the epicardium and endocardium, mucosal surface of the jejunum and serosal surface of the spleen; petechial haemorrhage of the pancreas and dorsal peritoneal surface, and haemorrhagic consolidation of the lung lobes. Of notable difference to historic cases is marked submandibular lymphadenomegaly (measuring approximately 10 cm x 7 cm x 7 cm). Histopathological findings were limited by post-mortem autolysis with a concentration of post-mortem clostridial-type bacteria in blood vessels creating significant artifact. Histopathological abnormalities included widespread perivascular haemorrhage, most prominent in the heart and tongue, and lymphoid depletion of lymph nodes and spleen. Vasculitis was present in the rectal and intestinal submucosa but not identified in other tissues. No viral inclusion bodies were observed in any tissue, though visualisation may have been impeded by the degree of post-mortem autolysis.

## Viral load highest in heart tissue, best preserved in viral transport medium

Elephant DNA and EEHV1B viral DNA was detected in all samples tested (Table 2). The highest amount of normalised viral DNA was detected in heart tissue, followed by the liver, gastrointestinal tract and tongue (Fig 2). Lower amounts of normalised viral DNA were detected in kidney samples. For heart tissue, storage in DNA/RNA Shield™ and viral transport medium resulted in greater amounts of viral DNA detected by qPCR compared to tissue stored in RNA*Later*™ or tissue without storage media. In other tissues there was no consistent relationship between storage media and the amount of viral DNA detected when normalised to the host cell number. Tissue stored in viral transport medium consistently demonstrated greater amount of both viral and host DNA detected by qPCR, compared with the other storage media (Table 2).

## Whole genome sequence assembled from heart tissue shares highest nucleotide identity with EEHV1B Emelia

Metagenomic sequencing on DNA extracted from heart tissue provided abundant viral DNA for EEHV whole genome assembly (~5% of total reads). After trimming for removal of sequencing adapters and PHRED quality filtering, a total of 42 million paired reads were obtained, of 30–150 bp in length. Of these reads, 3.1 million mapped to the final generated consensus sequence, with an average depth of 1,849 reads per nucleotide. In comparison, only 140,600 reads of a total 39 million paired reads from intestinal tissue mapped to this final consensus, with an average depth of 134 reads per nucleotide. When comparing the genome assembly with viral reads obtained from intestinal tissue to heart tissue, 14 nucleotide variants were identified (when excluding the terminal repeat regions and variants with read coverage <10) with eight of these within coding regions. Variants within coding sequences included a frame-shift insertion of E9 gene

**Table 2. Calculated quantity of EEHV1B virus (EEHV copies/µL) and host DNA (TNF-$\alpha$ copies/µL) in different tissue types stored without preservation media, and stored in DNA/RNA Shield, RNA*Later* and viral transport medium.**

|  | Plain (no media) | | DNA/RNA Shield | | RNA*Later* | | Viral transport medium | |
|---|---|---|---|---|---|---|---|---|
|  | EEHV1 DNA | Host DNA | EEHV1 DNA | Host DNA | EEHV1 DNA | Host DNA | EEHV1 DNA | Host DNA |
| **Brain** | <*LOD*[a] | $7.18 \times 10^2 \pm 8.8$ | <*LOD* | $8.96 \times 10^3 \pm 1.21 \times 10^3$ | N/A[b] | N/A | $1.14 \times 10^2 \pm 56.3$ | $4.05 \times 10^3 \pm 7.80 \times 10^2$ |
| **Bronchial lymph node** | $1.32 \times 10^3 \pm 4.80 \times 10^2$ | $50.2 \pm 28.9$ | $1.07 \times 10^3 \pm 32.6$ | $6.00 \times 10^2 \pm 33.3$ | N/A | N/A | $3.25 \times 10^4 \pm 1.45 \times 10^4$ | $3.55 \times 10^4 \pm 4.56 \times 10^3$ |
| **Heart** | $2.64 \times 10^5 \pm 8.63 \times 10^4$ | $4.61 \times 10^2 \pm 56.9$ | $8.38 \times 10^5 \pm 1.24 \times 10^5$ | $1.73 \times 10^2 \pm 72.0$ | $7.46 \times 10^5 \pm 4.84 \times 10^4$ | $1.30 \times 10^3 \pm 80.7$ | $3.24 \times 10^6 \pm 7.85 \times 10^5$ | $2.64 \times 10^2 \pm 35.0$ |
| **Intestine** | $7.46 \times 10^2 \pm 2.06 \times 10^2$ | $2.22 \times 10^4 \pm 1.32 \times 10^3$ | N/A | N/A | N/A | N/A | $6.66 \times 10^4 \pm 7.65 \times 10^3$ | $1.65 \times 10^4 \pm 1.03 \times 10^3$ |
| **Kidney** | $1.00 \times 10^3 \pm 41.1$ | $8.80 \times 10^4 \pm 2.01 \times 10^3$ | $4.35 \times 10^2 \pm 15.1$ | $8.64 \times 10^4 \pm 7.00 \times 10^3$ | $3.18 \times 10^3 \pm 9.21 \times 10^2$ | $1.67 \times 10^2 \pm 11.2$ | $7.31 \times 10^3 \pm 2.33 \times 10^3$ | $2.66 \times 10^5 \pm 7.47 \times 10^3$ |
| **Liver** | $1.07 \times 10^4 \pm 3.79 \times 10^3$ | $2.47 \times 10^2 \pm 16.4$ | $1.80 \times 10^3 \pm 1.78 \times 10^2$ | $22.6 \pm 3.4$ | $3.29 \times 10^3 \pm 69.6$ | $4.12 \times 10^4 \pm 8.09 \times 10^3$ | $7.53 \times 10^3 \pm 7.93 \times 10^2$ | $3.07 \times 10^2 \pm 25.5$ |
| **Lung** | $1.84 \times 10^4 \pm 1.78 \times 10^3$ | $1.53 \times 10^5 \pm 1.63 \times 10^2$ | $1.44 \times 10^4 \pm 4.20 \times 10^2$ | $1.59 \times 10^5 \pm 1.44 \times 10^3$ | N/A | N/A | $2.70 \times 10^4 \pm 3.41 \times 10^3$ | $2.34 \times 10^5 \pm 6.39 \times 10^3$ |
| **Spleen** | $3.61 \times 10^3 \pm 1.32 \times 10^3$ | $6.68 \times 10^4 \pm 6.93 \times 10^2$ | $9.14 \times 10^2 \pm 2.60 \times 10^2$ | $4.17 \times 10^3 \pm 1.26 \times 10^2$ | N/A | N/A | $4.44 \times 10^3 \pm 1.10 \times 10^3$ | $5.30 \times 10^4 \pm 4.27 \times 10^3$ |
| **Stomach** | $5.06e \times 10^3 \pm 1.68 \times 10^3$ | $1.48 \times 10^2 \pm 1.3$ | N/A | N/A | $3.58 \times 10^4 \pm 8.40 \times 10^3$ | $9.98 \times 10^3 \pm 2.78 \times 10^2$ | $1.14 \times 10^5 \pm 2.09 \times 10^4$ | $43.6 \pm 5.9$ |
| **Tongue** | N/A | N/A | $1.41 \times 10^4 \pm 2.15 \times 10^3$ | $6.16 \times 10^3 \pm 8.83 \times 10^2$ | N/A | N/A | $1.11 \times 10^5 \pm 2.35 \times 10^4$ | $3.42 \times 10^4 \pm 2.34 \times 10^2$ |

[a] <*LOD* indicates where calculated viral quantity was lower than the limit of detection, [b] N/A indicates tissue was not stored in that media and thus not available for testing.

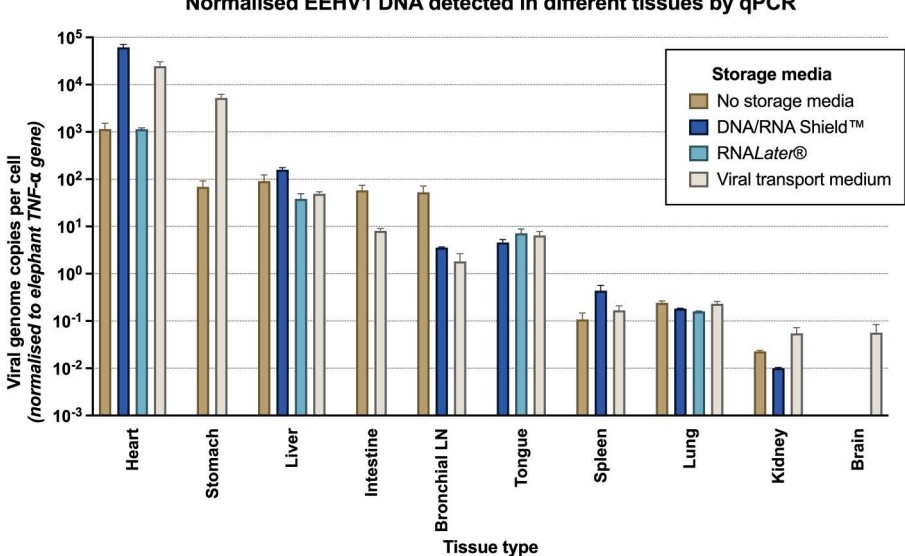

**Fig 2. EEHV1B viral genome copies (and standard deviation) in different tissues and storage media, normalised to number of cells (as determined by qPCR copy number of host reference gene TNF-$\alpha$).** Viral quantities were determined by EEHV1 qPCR, and all EEHV1 positives were subsequently subtyped to EEHV1B.

(a membrane protein) and multiple insertions and variants in the E34 gene (major tegument protein). Given the greater read depth, the following genomic analysis was performed using viral DNA from heart tissue.

The method of assembly described had superior read coverage, depth and final sequence length compared with other techniques trialled (see S4 Table). The final genome sequence of AUP_01 (GenBank accession: PX651398) had a total length of 178.3 kb, with 42.1% G + C nucleotide composition and encoded 118 open reading frames (ORFs) in the unique long region, flanked by terminal repeats of 2.83 kb. This is a similar size genome to EEHV1B Emelia (180 kb) and EEHV1A genomes (177 kb for Kimba NAP23, 180 kb for Raman/KC462165) and similar GC content (EEHV1B: 42.1% EEHV1A: 42.2% − 42.3%) [18,29]. Whole genome alignment to EEHV1B Emelia revealed a 96.0% pairwise nucleotide identity, whereas identity to EEHV1A reference sequences Kimba and Raman was 89.3% and 90.9% respectively. Across core conserved herpesvirus genes, nucleotide identity to EEHV1B was 99.8%, and 92.3% to EEHV1A. Phylogenetic analysis comparing AUP_01 to other EEHVs showed that AUP_01 clusters closely with EEHV1B (Fig 3).

There were 3,102 nucleotide variants detected compared with the EEHV1B reference sequence, 682 of which were excluded from analysis due to position within the terminal repeat or R2 cassette region. Of the remaining 2,420 variants, 905 were within non-coding regions and 1,515 were in annotated coding sequences. The transition to transversion SNP ratio across the genome was 2.18, and in coding sequences 2.67. The ratio of non-synonymous to synonymous substitutions (dN/dS) showed high purifying selection pressure with 85% of CDS analysed demonstrating dN/dS of < 0.5 (Fig 4). Only five genes showed dN/dS > 1.0, suggestive of diversifying selection, including E18A, E18B, E31, E37 and E38 genes. The frequency of SNPs varied across the genome (Fig 5) and was lowest in the conserved core regions C1 and C2 (mean SNPs: 8.8, 6.3 per CDS respectively), and highest in the R2, L2 and L1 regions (mean SNP's: 43.8, 19.7, 18.5 per CDS respectively). Some coding sequences of AUP_01, including genes E4, E67 contained frameshifts or premature truncations resulting in pseudo-genes.

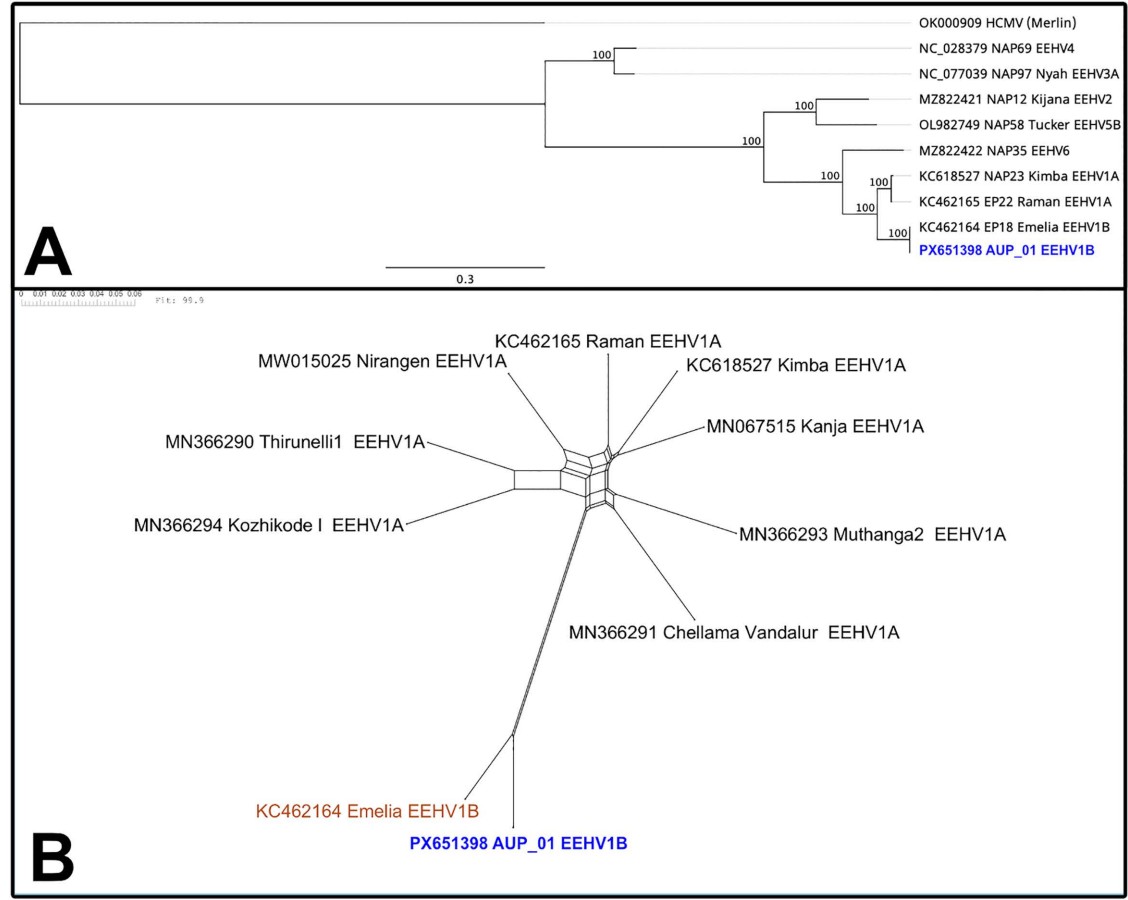

**Fig 3. Phylogenetic analysis of AUP_01 (in blue) compared to previously published EEHV sequences. (A)** Maximum likelihood phylogenetic tree generated by PhyML based on MAFFT alignment of six core-conserved herpesvirus genes (DNA polymerase [U38], glycoprotein B [U39], major capsid protein [U57], terminase [U60], helicase [U77] and uracil DNA-glycosylase [U81]), concatenated using the reference sequence of each EEHV species, **(B)** SplitsTree recombination network tree from alignment of whole genome of AUP_01 to other EEHV1 whole genome sequences, p-value determined by pairwise homoplasy inference (PHI) test for statistically significant recombination.

## Detection of EEHV1B and EEHV1A recombination in strain AUP_01

Pairwise whole genome alignment with EEHV1B (Fig 6A) showed that while nucleotide identity of AUP_01 to KC462164 is mostly consistent across the entire genome, there are several regions of lower sequence identity, correlating with genome regions L1, L2 and R2. This is supported by the SimPlot analyses of the whole genome sequence (Fig 6B) and partial genome sequences where AUP_01 shares greater identity with multiple different EEHV1A sequences in these distinct regions (Fig 7A–7C). These SimPlots demonstrate sharp transition zones of dissimilarity to the EEHV1B reference sequence, while maintaining a high degree of similarity to one or more EEHV1A sequences. The genes of AUP_01 which shared the greatest nucleotide identity to EEHV1A were E3, E5, E24, E25 and the entire R2 region. Across the 9.8kb R2 region, this sequence shared >99% nucleotide identity to EEHV1A sequence NAP20 KathyShboom (MH287527) and only 68% identity to EEHV1B Emelia.

Recombination network trees (i.e., SplitsTrees) demonstrated complex reticulate networks and PHI test for recombination of $p < 0.001$ supporting evidence of past recombination events between AUP_01 and EEHV1A and EEHV1B sequences at these regions of interest (Figs 8A-C). These regions showed EEHV1A and EEHV1B sequences clustering

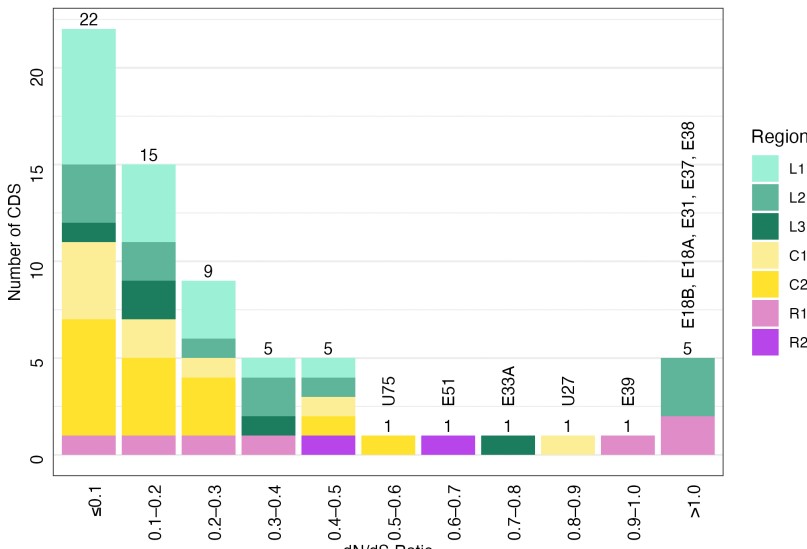

**Fig 4. The number of coding DNA sequences (CDS) categorised by the ratio of non-synonymous (dN) to synonymous substitutions(dS).** A dN/dS ratio <1.0 represents negative/purifying selection against non-synonymous substitutions, dN/dS = 1 represents neutral genetic drift and dN/dS > 1 represents positive diversifying pressure. Stacked columns are further identified by region of the genome in which the CDS is located.

into clades independently of their subspecies designation. SplitsTree analysis at the C1/C2 region and across the entire genome (Figs 8D, Fig 6B) show less obvious reticulate networks between subspecies, though still evident within the EEHV1A strains.

## Discussion

This report describes EEHV1B infection as a cause of fatal haemorrhagic disease in a nine-year-old male Asian elephant in Australia. Primary infections with EEHV are increasingly recognised as a conservation threat, primarily affecting young elephants in *ex situ* conservation herds [6,13]. Fatal infection with EEHV1B is uncommon compared with EEHV1A, with only 13 fatal cases of EEHV-HD caused by EEHV1B previously described [5,7,10,17,60]. This report highlights that age alone may not reflect individual risk of EEHV-HD, that viral shedding in trunk washes increased within the herd following a fatal event and that viral transport media best preserves virus in infected tissue. Additionally, this study revealed that recombination between EEHV1A and EEHV1B subspecies drove variability within an otherwise well conserved genome.

Fatal EEHV-HD infection in a nine-year-old elephant is very uncommon, with less than ten cases previously reported in the literature in animals over nine years of age [4,6,11]. Protective antibodies tend to increase with age, with 95% of elephants older than 10 years of age having EEHV-specific antibodies at levels considered protective [61]. This correlation between age and antibodies is likely due to EEHV exposure risk accruing with age and cycles of lytic infection due to reactivation [61]. Disease occurrence in older elephants is likely caused by delayed novel exposure to a species of EEHV, which may occur due to prolonged periods of latency in herd-mates, small herd sizes resulting in less potential for exposure, or older contact with new elephants latently infected with novel EEHVs [13,14,26]. Recent studies suggest that serology is a reliable indicator of EEHV-HD risk and therefore should be considered as part of routine monitoring and management of EEHV-HD [14,61]. Serological assays were not available in Australia at the time of this case and thus could not be used to inform disease risk, resulting in the serostatus of the affected animal and that of the other herd members to be unknown. The development of regional serology testing capacity would inform risk-based sampling frequency and thus enhance EEHV-HD surveillance.

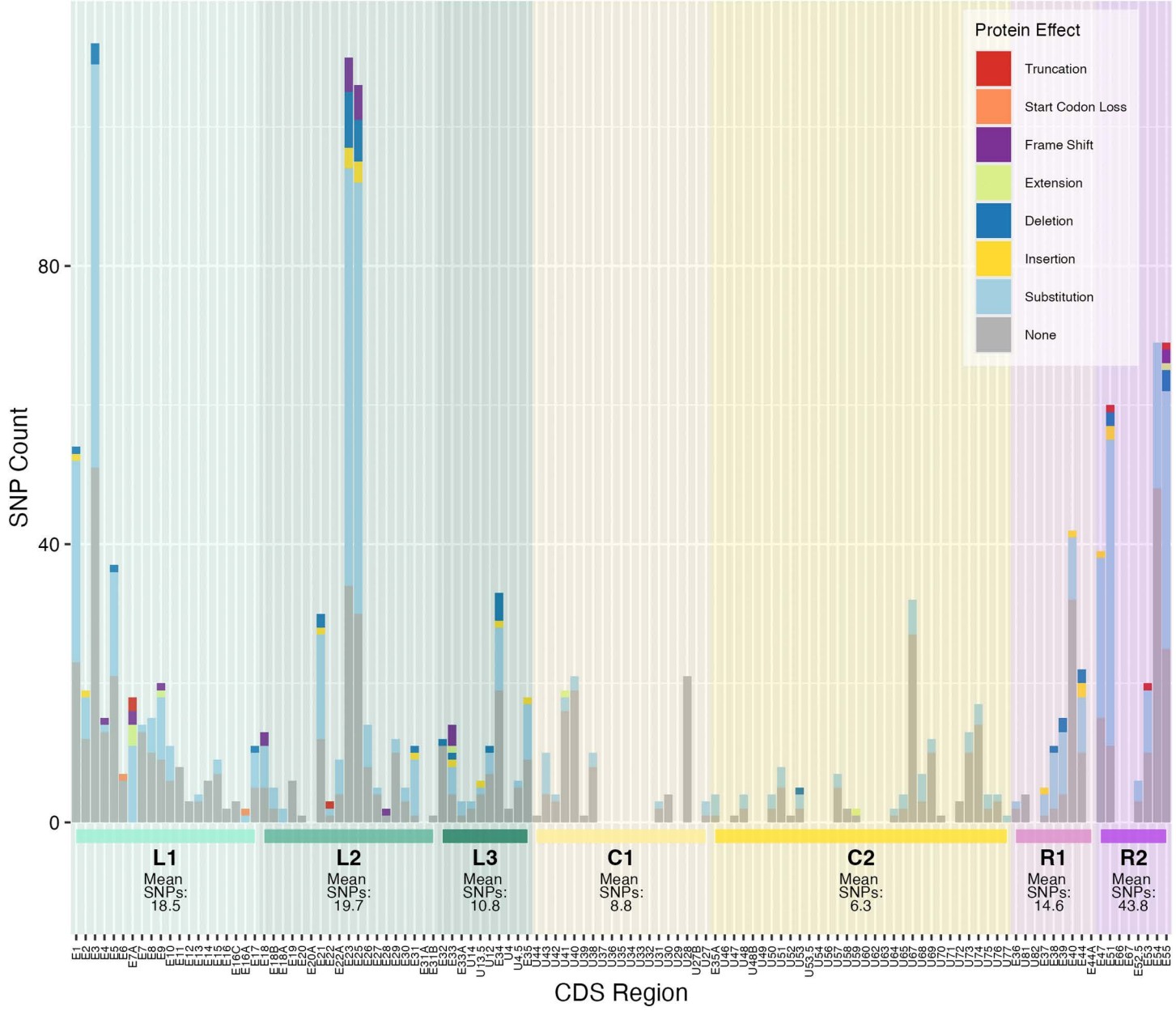

**Fig 5. The number of polymorphisms per CDS site.** Polymorphisms are characterised by protein effect (bar colours) and identified by genome region (shading). Mean SNPs per genome region are calculated by averaging SNPs per CDS for each gene region.

This case shares parallels to a previous case report of two non-fatal EEHV1B cases [62]. In that study, initial viraemias were detected at a much lower viral load (<10,000 VGE/mL), which likely indicates earlier infection status. Subsequently anti-viral and supportive therapies were therefore likely initiated earlier in the clinical progression of infection. In the second case described by this prior study, viraemia rapidly rose from $16.5 \times 10^3$ VGE/mL to $2.13 \times 10^5$ VGE/mL within five days and peaked at $4.53 \times 10^5$ VGE/mL after eleven days. Treatment in the cases described by Fuery et al. (2016) followed current therapeutic recommendations and thus was mostly consistent with the case described in our study.

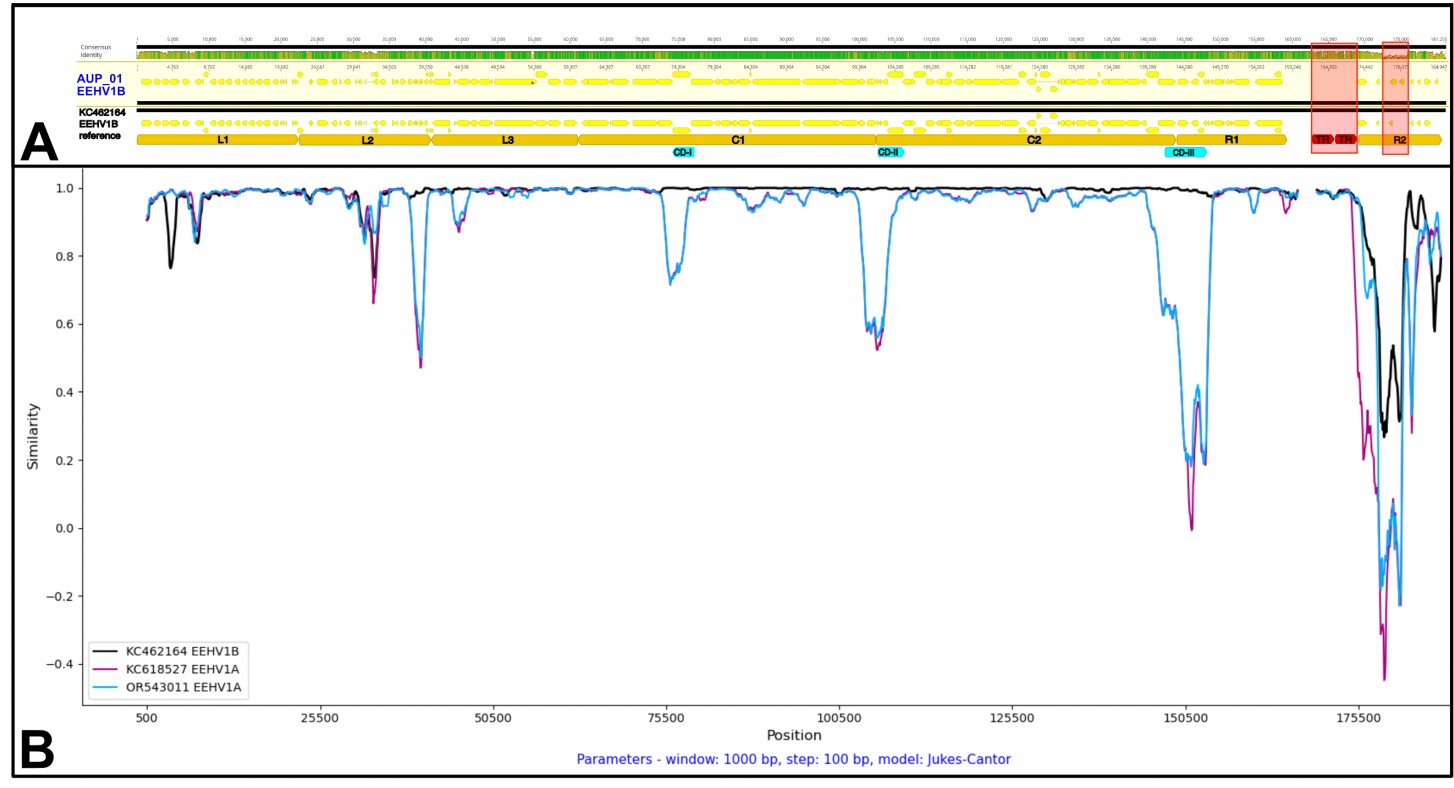

**Fig 6. Whole genome alignment and similarity plot of AUP_01 to EEHV1B Emelia reference sequence. (A)** Geneious pairwise alignment and genome map demonstrating nucleotide identity (top bar), CDS coding regions (yellow), chimeric domains CDI-III (blue), genome region (orange), terminal repeat regions (red) and regions excluded from SNP analysis (red box overlay); **(B)** SimPlots of EEHV1B Emelia (KC462164: black) and EEHV1A reference sequences (KC618527: magenta, OR543011: blue) with EEHV1B_AUP_01_2023 as the query.

Difference in clinical outcomes between cases may reflect differences in EEHV1B strain virulence, individual immune competence or preclinical detection of virus.

The epidemiology of EEHV infection spread within a herd is still poorly understood but surveillance of trunk wash samples for EEHV shedding, along with detection of EEHV in whole blood sampling, can assist with understanding viral infection dynamics and thus management of EEHV-HD risk [24,26,63]. Trunk wash sampling has been demonstrated to be more sensitive for detection of EEHV, compared to oral, conjunctival and urogenital swabs, and to that of urine and faecal testing [64–66]. In this study, peak EEHV1B shedding occurred seven weeks prior to the onset of disease, with only low levels of virus detected in trunk washes two weeks prior to the onset of disease. This may suggest a prolonged incubation period greater than the two weeks that has been previously suggested [26], or could represent delayed exposure through fomite transmission such as contaminated shared water source as is seen with other herpesviruses [67]. Alternatively, increased viral shedding closer to the time of disease onset may not have coincided with the timing of trunk wash sampling. Trunk wash data spanning nine months from Elephant 5 suggests that this method of fortnightly surveillance lacks sensitivity for EEHV shedding and increasing the frequency to weekly trunk wash surveillance, as is currently recommended by the EEHV advisory group, would likely increase detection of EEHV shedding events [68].

The pattern of EEHV1B detection in trunk washes of multiple herd members after the clinical case of EEHV1B, despite a lack of viraemia, may reflect a high level of environmental viral contamination from the affected animal or subclinical

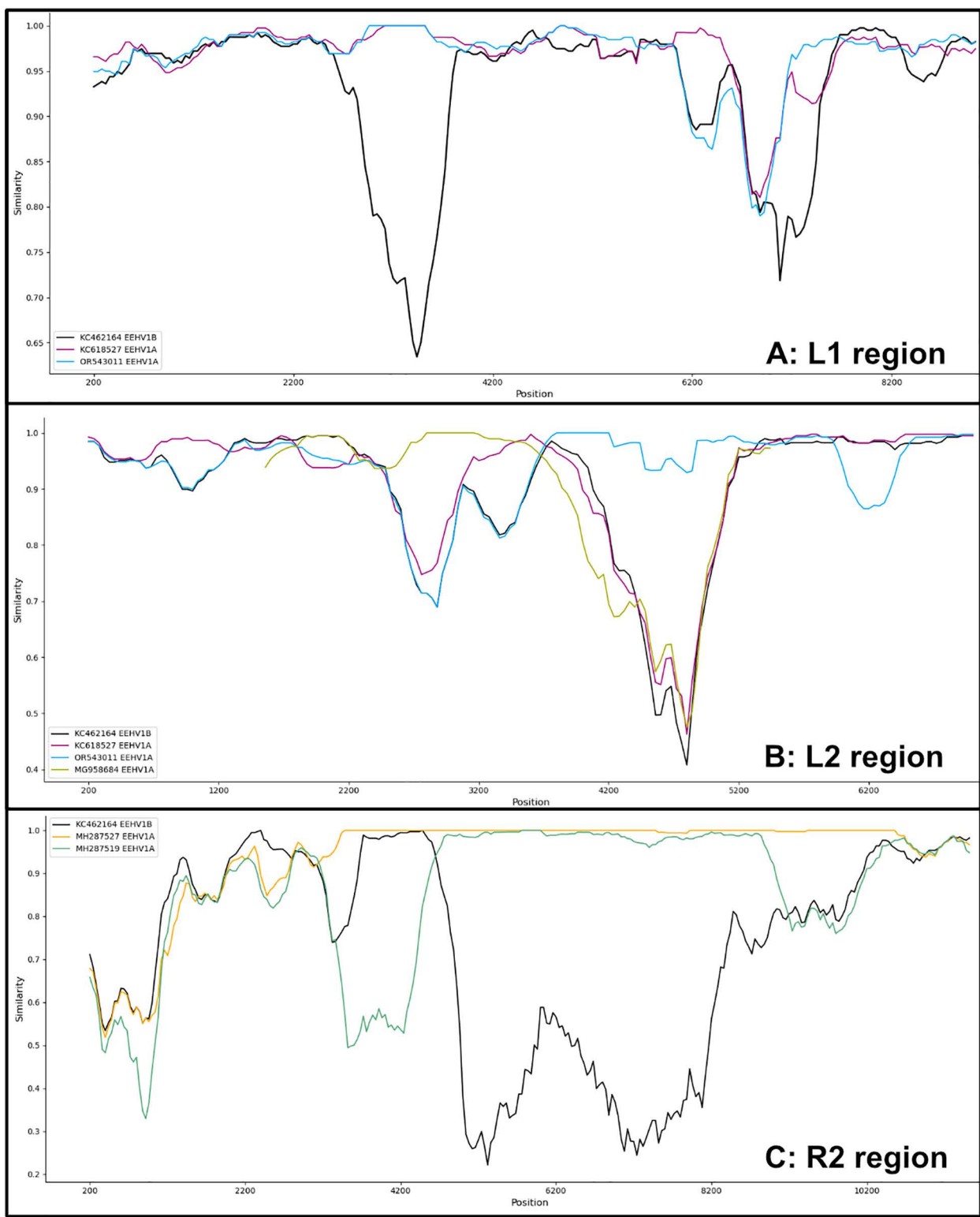

**Fig 7. SimPlots of genome regions displaying divergence of AUP_01 sequence to the EEHV1B reference genome. (A)** L1 region (E1 – E7, 9.2kb); **(B)** L2 region (E21 – E26, 7.2kb); **(C)** R2 region (E47 – E55, 9.8kb). Genomes displayed are the EEHV1B Emelia reference genome (KC462164: black), and EEHV1A sequences (KC618527: magenta, OR543011: cyan, MG958684: olive, MH287527: gold, MH287519: green) demonstrating highest nucleotide identity to AUP_01.

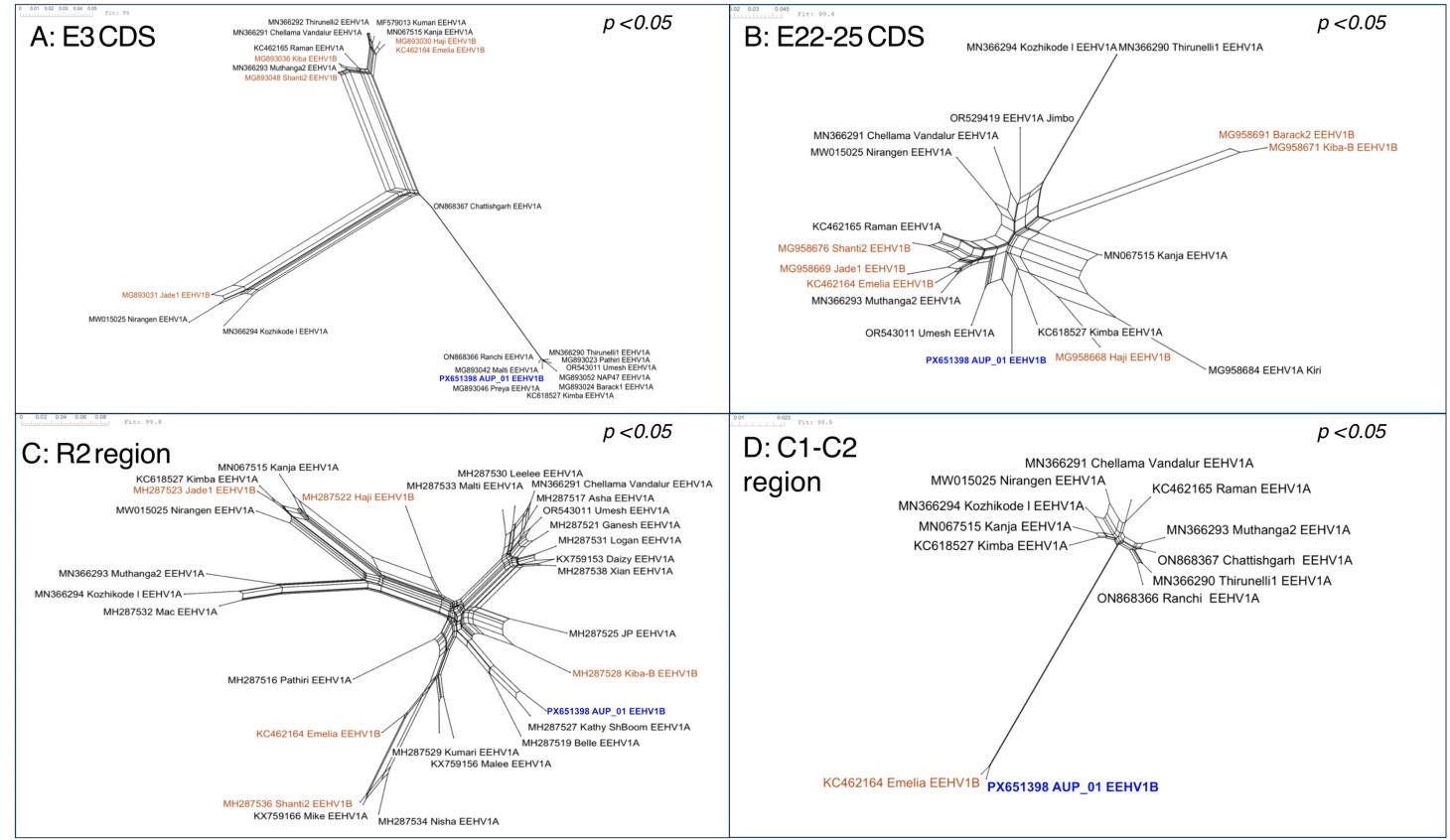

**Fig 8. SplitsTrees recombination network trees generated from alignment of AUP_01 (blue, bolded) to EEHV1A (black) and EEHV1B (brown) GenBank published sequences. (A)** L1 region (E3 CDS, 1 kb); **(B)** L2 region (E22 – E25 CDS, 5 kb); **(C)** R2 region (E47 – E55, 9.8 kb); and **(D)** Core conserved regions C1 – C2 (82.9 kb). Statistical significance (*p*) determined by pairwise homoplasy inference (PHI) test for recombination.

infections within the herd. Importantly, EEHV1B shedding in trunk wash samples of herd-mates before and following a clinical case of EEHV1B disease has not been described previously, and so the data presented in this report therefore contributes important new information to our understanding of EEHV1B infection and transmission dynamics. Prior EEHV1B clinical cases described trunk wash shedding in two affected elephants up to three months after viraemia, but did not describe trunk wash detection rates across the remainder of the herd [62]. A large cross-sectional study of 10 European zoos documented a prevalence of EEHV1 in trunk washes as 13.7% (391/2,863) [66]. Comparatively, the EEHV1B prevalence in this study was much higher (44.4%) in the three months following the clinical event, but more similar in the subsequent nine months (23.4%).

Pathology and histopathology findings were mostly consistent with previously described findings [8,28,69]. This case uniquely demonstrated marked enlargement of the submandibular lymph nodes, though oedema, lymphangiectasia (dilation of lymphatic vessels) and lymph node abscessation have previously been observed [8]. Lymphadenomegaly in this case may be attributed to a combination of localised inflammatory reaction and oedema. Additional inferences from histopathology were difficult due to the interval from death to post-mortem.

The highest viral loads were detected in heart tissue, and exceed those reported in a previous study that performed a comprehensive molecular investigation of viral load and tissue tropism in fatal EEHV1A and EEHV1B cases [59]. There they detected high levels of EEHV DNA in the heart consistent with this study, with EEHV1B DNA detection in the heart

approximately 57-fold higher than in the liver, and 30-fold higher than in the tongue. This is in contrast to EEHV1A cases where the highest viral loads were detected in the liver [59]. This prior study did not investigate gastrointestinal tissue viral loads, so our study newly documents potential tropism for stomach and intestinal tissues based on comparative viral loads to other tissues. Differences in viral load and tissue tropism between EEHV1B cases may be due to a difference in host responses to infection (including host immune status) and/or viral factors that may influence virus tropism, but this is currently unknown. Other studies have utilised *in situ* hybridisation [70,71] and immunohistochemistry [11,72] techniques targeting the POL and TER genes of EEHV1A infected tissues to explore tissue tropisms. These studies similarly demonstrated an affinity of EEHV1A virus to endothelial cells of the heart, spleen and liver as well as the tongue, more so than other tissues such as the kidneys, intestine and lung. Studies comparing the viral load of tissues in fatal EEHV infections have not previously explored the effect of storage media on recovery of viral DNA and have typically used formalin-preserved or cryopreserved tissues, or otherwise not specified means of tissue preservation [59,69,71]. The results from this study suggest that viral transport medium and DNA/RNA Shield are the preferred media for viral DNA recovery from tissue, however there was still adequate viral DNA from tissues stored at −80 °C without media for molecular detection assays. In resource-limited post-mortem scenarios, it would be sufficient to collect tissue samples for qPCR and store them frozen without media.

The generated sequence, AUP_01 (GenBank accession: PX651398), represents the second EEHV1B whole genome sequence published and is arranged in parallel to the EEHV1A reference sequences published by Richman, Zong [17], Ling, Reid [29]. The genomic composition as inferred from GC content and genome size are consistent with published EEHV1A and EEHV1B reference sequences [18]. Comparison of AUP_01 to the EEHV1B Emelia reference genome identified a ratio of transition to transversion substitutions similar to that documented in other *Betaherpesvirinae* (1.81–2.80) [43]. This high transition/transversion ratio exceeds that of previously examined alphaherpesviruses and gammaherpesviruses, suggesting that betaherpesviruses are more host adapted and most genes have a strong purifying selection, removing transversions which result in more non-synonymous substitutions [43]. Another measure reflecting this high degree of purifying selection is the proportion of genes displaying a dN/dS ratio of less than 1.0. Between AUP_01 and the EEHV1B Emelia, less than 5% of genes demonstrated a diversifying selection pressure (dN/dS > 1.0) and these genes included signalling proteins which may be under immune selection pressure from the host.

There were multiple distinct regions in AUP_01 where similarity switched from EEHV1B to EEHV1A reference genomes, indicative of recombination. This includes the E3 (vGPCR6), E5 (vGPCR5), E23-25 (vOX2–2/3/4) and the R2 regions. Previous work identified these regions as hypervariable regions that show subtype grouping which is independent of EEHV1A/1B subspeciation [14,24,27,28]. The R2 region contains six hypervariable genes and at least eight alternative groups of inserted 'triplex gene cassettes' which each have different gene contents and insertion points [27]. Importantly, each of these hypervariable regions form subgroups unique to each specific gene site, with a mosaic pattern of subgroups across an individual EEHV genome [27,28]. Despite recognition of these hypervariable regions, the mechanism of genomic variation in EEHVs still remains poorly understood. When comparing the genome derived from intestinal tissue to that of heart tissue, 14% of SNPs (n = 2/14) were located within these hypervariable regions, therefore this hypervariability is not apparently driven by a rapid mutation rate. Recombination in EEHV1 has been described primarily as an ancient event, resulting in chimeric domain region differences between EEHV1A and EEHV1B [17], however the role of recombination between EEHV sub-species and strains remains relatively unexplored compared with other herpesviruses [43,73,74].

The recombination analysis performed in this study suggests that recombination drives variability in these hypervariable regions. Recombination network trees demonstrated complex reticulate networks which suggest conflicting evolutionary signals (i.e., recombination) at each of the sites examined. Furthermore, the distribution of EEHV1A and 1B sequences within these trees was independent of sub-species, with AUP_01 clustering in proximity to EEHV1A sequences for each of the examined hypervariable sites. Previous multi-locus sequencing work that investigated eight PCR loci in 13 EEHV1

sequences drew similar conclusions about evidence of recombination [28]. This previous study identified regional distribution between subtype patterns at particular PCR loci including U48 (gH-TK), U51 (vGPCR1), E5 (vGPCR5) and E54 (vOX2−1) genes, where strains from elephants in India and Thailand differed from each other and the subtype patterns in strains from elephants in Europea and North America. However geographical subtype patterns were localised to individual PCR loci and were unlinked to adjacent loci. The limitation of multi-locus sequencing in contrast with whole genome sequencing is the lack of detail about genomic conservation and recombination in other sites. Further supporting evidence for recombination in EEHV1 is the recently described 'gene cassette' nature of the R2 region [27] in which the described exchange of genes between EEHVs may only occur by genetic recombination. Lastly, recombination is possible at the chimeric domain regions with at least one case (NAP#19 "Haji") displaying a pattern of EEHV1B-1A-1B across the three chimeric domains [17]. Analysis of the conserved regions C1-C2 in this study showed evidence of recombination between EEHV1A strains, but not between 1A and 1B subspecies. This likely reflects that recombination events occur more commonly at either genomic termini, as has been demonstrated in other herpesviruses [73]. Visualising the effect of recombination on the whole genome alignment is impeded by the fact that the EEHV1A sequences that have the highest similarity to AUP_01 are only available as partial genome sequences and therefore unable to be included for whole genome analyses.

The significance of recombination can be explored by examining the proposed function of genes in these recombinant regions. The E3 and E5 genes both encode for viral G-protein coupled receptors (vGPCRs) [29], which play several roles such as ligand-induced signalling, promotion of cell growth and survival or enhanced cellular entry [75]. In other herpesviruses, these vGPCRs are capable of rewiring host cellular signalling networks resulting in secretion of growth factors and cytokines and are hypothesised to contribute to viral escape of host immune surveillance [43,76–78]. The E23, E24, E25 genes are three captured cellular OX2-like membrane proteins [31]. Viral OX2-like proteins interact with CD200 receptors on myeloid cells and may play a role in dampening monocyte and macrophage-mediated responses to infection [79,80]. The R2 region encodes other vGPCRs and vOX2-like proteins, as well as membrane-bound immunoglobulin domain proteins (including E66 and E67 in the inserted cassette region of this genome) and a fucosyltransferase protein (vFUT9) [31]. Immunoglobulin domain proteins in viruses play a number of roles in immune evasion mechanisms through ligand-binding interactions, including induction of T-cell and monocyte migration, and downregulation of the adaptive immune response [81]. Each of these hypervariable regions therefore may play an important role in the pathogenesis of infection, and recombination may be considered a mechanism of driving immune-evasion or host adaptation. Transcriptomic analysis of these genes through various time-points of infection may provide further evidence to support this theory, however the absence of a successful viral cultivation model impedes such studies [6,82]. Recombination between EEHV1A and EEHV1B at the chimeric domain sites as described by Richman, Zong [17] may be of greater clinical concern, particularly at sites targeted by molecular detection assays (U41 - EEHV1, U71, U48.5 - EEHV1A/B subtype) [23,25], glycoprotein vaccine targets (U38 - gB, U48 - gH, U82 - gL) [83], and candidate anti-viral targets (U48.5 - thymidine kinase) [27]. Such recombination events may render current diagnostics, vaccines and therapeutics inaccurate or ineffective for a given EEHV strain. There is a relative lack of whole genome data to sufficiently analyse the possibility and frequency of such recombination, and therefore ongoing whole genome sequencing should be undertaken.

Compared with other viruses, herpesviruses have a low mutation rate due to high-fidelity DNA replication and efficient DNA repair mechanisms [84,85]. Genomic analysis of several other herpesviruses has revealed widespread recombination, and this has been hypothesized as an important component of herpesvirus evolution in the face of such a low mutation rate [86,87]. There is a positive correlation between genomic variation and recombination in several herpesviruses including diverse equine herpesviruses (EHV) [74,88], avian infectious laryngotracheitis virus (ILTV) [79,80] and herpes simplex virus type 1 (HSV1) [89]. The role of recombination has also been explored in several betaherpesviruses, including human cytomegalovirus (HCMV) [43,76–78] and murine cytomegalovirus (MCMV) [73]. In a study of recombination in MCMV, 47% of genes (n = 62/131) demonstrated statistically significant evidence of recombination between 11 genomes

[73]. HCMV studies have shown that while most of the genome is highly conserved, there is pervasive recombination which is linked to the common occurrence of co-infections [28]. Given the lack of cross-protective immunity between EEHV subspecies [14], and the persisting nature of herpesvirus infections [90], there is probable opportunity for recombination between EEHV subspecies. Similarly, co-infection with different EEHV species has been documented in both fatal disease [91] and detected incidentally in screening tests [92], and therefore could represent opportunity for interspecies recombination. However, the greater nucleotide divergence between EEHV species, particularly the highly diverged GC-rich branch (EEHV-4), reduces the likelihood of recombination between these EEHV species [31]. This study contributes to our understanding of EEHV genomics, which provides further insights into their evolution, patterns of recombination, as well as to the pathophysiology of EEHV infection.

## Conclusions

This case-study documents a fatal case of EEHV-HD due to infection with EEHV1B in an Australian Zoo and outlines the EEHV herd viral dynamics and tissue tropism of infection. Next generation sequencing of viral DNA from infected tissues contributes important information to our understanding of EEHV genomics, with further evidence provided for the occurrence of recombination between EEHV1A and EEHV1B subspecies. With relatively few cases of disease due to EEHV1B described in the scientific literature, careful documentation of future cases of EEHV1B infection is critical to better understand the pathogenesis of this disease and inform disease control and management programs. This may require an ongoing partnership between zoological institutions that care for these animals in captivity and research institutes that have the capacity for such analysis.

## Supporting information

**S1 Table. Demographics of elephants within the Zoos Victoria herd at the time of the EEHV-HD clinical event.** There were no clinical samples available from Elephants 8–10 as they were in training for voluntary sample collection. (XLSX)

**S2 Table. A list of all genes identified within the genome EEHV1B_AUP_01_2023 based on the transfer of annotations from existing EEHV genomes.** Disambiguation of gene names is provided to clarify the two different naming conventions, as well as information regarding position within genome, proposed genome function (as defined by genomes used for annotation transfer), and identity between EEHV1B_AUP_01_2023 and EEHV1B reference genomes and closest match EEHV1A genome. (XLSX)

**S3 Table. List of all EEHV genomes used in this analysis.** Information is provided regarding the geographical location and year of sample collection, the sample ID and GenBank accession data, the total genome length and the reference described where available. (XLSX)

**S4 Table. Comparison of genome assembly methods.** The total number of reads mapped, average depth of coverage and length of final genome is presented for different assembly methods. (XLSX)

## Acknowledgments

The authors acknowledge the veterinary department and the elephant keeper team for their dedicated efforts to the treatment of this elephant and the collection of diagnostic samples, as well as the Melbourne Veterinary School Anatomic Pathology team for provision of tissue samples.

                                                              

## Author contributions

**Conceptualization:** Jack W. Wheelahan, Paola K. Vaz, Alistair R. Legione, Carol A. Hartley, Natalie L. Rourke, Michael Lynch, Bonnie McMeekin, Joanne M. Devlin.

**Data curation:** Jack W. Wheelahan, Alistair R. Legione.

**Formal analysis:** Jack W. Wheelahan.

**Funding acquisition:** Jack W. Wheelahan, Joanne M. Devlin.

**Investigation:** Jack W. Wheelahan.

**Methodology:** Jack W. Wheelahan, Paola K. Vaz, Alistair R. Legione.

**Project administration:** Jack W. Wheelahan, Joanne M. Devlin.

**Resources:** Natalie L. Rourke, Michael Lynch, Bonnie McMeekin, Elizabeth C. Dobson.

**Supervision:** Paola K. Vaz, Alistair R. Legione, Carol A. Hartley, Natalie L. Rourke, Joanne M. Devlin.

**Validation:** Paola K. Vaz.

**Writing – original draft:** Jack W. Wheelahan.

**Writing – review & editing:** Paola K. Vaz, Alistair R. Legione, Carol A. Hartley, Natalie L. Rourke, Elizabeth C. Dobson, Joanne M. Devlin.

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
