## [Decision Letter · Decision Letter 0]

3 Apr 2026

PONE-D-26-06285Virological investigation of elephant endotheliotropic herpesvirus 1B infection in an Australian captive herd of Asian elephants (*Elephas maximus*)PLOS One

Dear Dr. Wheelahan,

Thank you for submitting your manuscript to PLOS ONE. After careful consideration, we feel that it has merit but does not fully meet PLOS ONE’s publication criteria as it currently stands. Therefore, we invite you to submit a revised version of the manuscript that addresses the points raised during the review process.

This article is clear and the information is interesting, but some minor improvment can be done as suggested by the two reviewers to clarify some clinical information.

We look forward to receiving your revised manuscript.

Kind regards,

Pierre Roques, Ph.D.

Academic Editor

PLOS One

Journal Requirements:

2. To comply with PLOS One submissions requirements, in your Methods section, please provide additional information regarding the experiments involving animals and ensure you have included details on (1) methods of sacrifice, (2) methods of anesthesia and/or analgesia, and (3) efforts to alleviate suffering.

This project was funded by Cybec Foundation, the Australian Research Training Program Grant (JW) and the Ernie Stewart Memorial Grant (JW). The authors acknowledge the veterinary department and the elephant keeper team for their dedicated efforts to the treatment of this elephant and the collection of diagnostic samples, as well as the Melbourne Veterinary School Anatomic Pathology team for provision of tissue samples.

This study was funded by generous donation from the Cybec Foundation to the University of Melbourne (J.D., J.W.) (https://www.cybec.org/), by the Australian Research Training Program Scholarship (Stipend) (J.W.), Australian Government (doi.org/10.82133/C42F-K220), and by the Ernie Stewart Memorial Scholarship (J.W.), University of Melbourne (https://scholarships.unimelb.edu.au/awards/ernie-stewart-memorial-scholarship). Funding sources were not involved in study design, collection, analysis or interpretation of data, nor writing of the report.

7. Please amend your authorship list in your manuscript file to include author Jack William Wheelahan.

8. Please amend the manuscript submission data (via Edit Submission) to include author Jack W. Wheelahan.

9. Please amend your list of authors on the manuscript to ensure that each author is linked to an affiliation. Authors’ affiliations should reflect the institution where the work was done (if authors moved subsequently, you can also list the new affiliation stating “current affiliation:….” as necessary).

10. We note that there is identifying data in the Supporting Information file <S1_file.xlsx>. Due to the inclusion of these potentially identifying data, we have removed this file from your file inventory. Prior to sharing human research participant data, authors should consult with an ethics committee to ensure data are shared in accordance with participant consent and all applicable local laws.

-Location data

Reviewers' comments:

Reviewer's Responses to Questions

**Comments to the Author**

1. Is the manuscript technically sound, and do the data support the conclusions?

Reviewer #1: Yes

Reviewer #2: Yes

2. Has the statistical analysis been performed appropriately and rigorously? 

Reviewer #1: Yes

Reviewer #2: Yes

3. Have the authors made all data underlying the findings in their manuscript fully available?

Reviewer #1: Yes

Reviewer #2: Yes

4. Is the manuscript presented in an intelligible fashion and written in standard English?

Reviewer #1: Yes

Reviewer #2: Yes

5. Review Comments to the Author

Reviewer #1: The manuscript by Wheelahan et al describes a lethal hemorrhagic disease case caused by EEHV1B in an elephant that resided in an Australian zoo. In addition to clinical findings, a complete genome sequence of EEHV1B was determined from heart tissue of the deceased animal. Few cases of EEHV1B have been described and correspondingly, there is little information available for the genomic landscape of EEHV1B strains or genotypes. Overall, this is a well written, comprehensive description of a rare case of EEHV1B-associated hemorrhagic disease and addition of another 1B genome sequence is an important contribution. While there is detailed discussion of finer aspects and peculiarities of the genome sequence, much of the speculation is not inherently novel above many of the observations put forth by several publications from the Hayward lab group.

Minor points:

1. page 24, line 516 “ age is an unreliable indicator”. This sentence is somewhat misleading and should modified. While the elephant in this case was just out of the normal range of observed EEHV HD cases, the age range 2-8 is still a predominant feature of EEHV HD. In addition, some discussion should be added as to why some elephants could be outliers from this age range.

2. page 25, lines 552-553, “EEHV1B shedding in trunk wash samples before and after a clinical case of EEHV1B disease has not been described previously…..”

EEHV1B trunk wash shedding has been described before the current study in Fuery et al, 2016, JZWM (Clinical Infection of Two Captive Asian Elephants (Elephas Maximus) With Elephant Endotheliotropic Herpesvirus 1b)

In fact, the authors should consider comparing observations from that study to the current one in either the intro or discussion or both.

Reviewer #2: The manuscript provides valuable information on the first reported case of EEHV1B infection in zoo Asian elephants in Australia, with important genetic insights that will interest the readership. However, several points require clarification or expansion to strengthen the manuscript. Please address the following comments to improve clarity, accuracy, and completeness.

Please see the attached file.

6. PLOS authors have the option to publish the peer review history of their article (what does this mean?). If published, this will include your full peer review and any attached files.

Reviewer #1: No

Reviewer #2: No

---

## [Author Response · Author response to Decision Letter 1]

24 Apr 2026

Dear editor/s and reviewers,

Thank you for your positive feedback and suggestions. Please find individual responses to each revision suggestion outlined below. Responses are indicated by blue text. Line numbers (L xx) for reference are those listed in the marked-up document. Please refer to the attached "Response to reviewer" doc for formatted responses.

Editor:

Response: The document has been formatted to the style guide, including the file naming convention.

2. To comply with PLOS One submissions requirements, in your Methods section, please provide additional information regarding the experiments involving animals and ensure you have included details on (1) methods of sacrifice, (2) methods of anesthesia and/or analgesia, and (3) efforts to alleviate suffering.

Response:

The study primarily encompasses post-mortem analysis of a clinical case of EEHV. All samples collected and treatments administered were for the purposes of clinical management of the case, independent of this report. No animal experimentation occurred within this study.

Response: A revised data availability statement is now added to the existing data availability statement as suggested: “All data are in the manuscript and/or supporting information files”.

5. We note that the grant information you provided in the ‘Funding Information’ and ‘Financial Disclosure’ sections do not match. When you resubmit, please ensure that you provide the correct grant numbers for the awards you received for your study in the ‘Funding Information’ section.

Response: Apologies for any incongruence in information listed. The detail listed below in the funding statement is correct:

“This study was funded by generous donation from the Cybec Foundation to the University of Melbourne (J.D., J.W.) (https://www.cybec.org/), by the Australian Research Training Program Scholarship (Stipend) (J.W.), Australian Government (doi.org/10.82133/C42F-K220), and by the Ernie Stewart Memorial Scholarship (J.W.), University of Melbourne (https://scholarships.unimelb.edu.au/awards/ernie-stewart-memorial-scholarship). Funding sources were not involved in study design, collection, analysis or interpretation of data, nor writing of the report.”

Upon resubmission, we will ensure this reflected in both the financial disclosure and funding information sections on resubmission. We do not have grant numbers available as the Cybec funding was a philanthropic gift bestowal to the University and the other two funding sources are scholarships.

This project was funded by Cybec Foundation, the Australian Research Training Program Grant (JW) and the Ernie Stewart Memorial Grant (JW). The authors acknowledge the veterinary department and the elephant keeper team for their dedicated efforts to the treatment of this elephant and the collection of diagnostic samples, as well as the Melbourne Veterinary School Anatomic Pathology team for provision of tissue samples.

We note that you have provided funding information that is not currently declared in your Funding Statement. However, funding information should not appear in the Acknowledgments section or other areas of your manuscript. We will only publish funding information present in the Funding Statement section of the online submission form. Please remove any funding-related text from the manuscript and let us know how you would like to update your Funding Statement. Currently, your Funding Statement reads as follows:

This study was funded by generous donation from the Cybec Foundation to the University of Melbourne (J.D., J.W.) (https://www.cybec.org/), by the Australian Research Training Program Scholarship (Stipend) (J.W.), Australian Government (doi.org/10.82133/C42F-K220), and by the Ernie Stewart Memorial Scholarship (J.W.), University of Melbourne (https://scholarships.unimelb.edu.au/awards/ernie-stewart-memorial-scholarship). Funding sources were not involved in study design, collection, analysis or interpretation of data, nor writing of the report.

Response: We have revised our acknowledgements (L792) to only identify non-financial acknowledgements, including the work of the vets and pathology teams involved in the case. All funding related text has been removed from acknowledgements.

7. Please amend your authorship list in your manuscript file to include author Jack William Wheelahan.

8. Please amend the manuscript submission data (via Edit Submission) to include author Jack W. Wheelahan.

Response: There appears to be a mismatch between the manuscript and submission authorship list, by inclusion of my full middle name. To clarify, they refer to the same individual, and I will edit the submission data to reflect the manuscript listed author to Jack W. Wheelahan.

9. Please amend your list of authors on the manuscript to ensure that each author is linked to an affiliation. Authors’ affiliations should reflect the institution where the work was done (if authors moved subsequently, you can also list the new affiliation stating “current affiliation:….” as necessary).

Response: I have reviewed this and I believe all authors had affiliations listed on the first submission, I cannot identify any errors compared with the submission guidelines. One co-author has requested I add an additional current affiliation. Please indicate if there are any issues requiring further attention.

10. We note that there is identifying data in the Supporting Information file <S1_file.xlsx>. Due to the inclusion of these potentially identifying data, we have removed this file from your file inventory. Prior to sharing human research participant data, authors should consult with an ethics committee to ensure data are shared in accordance with participant consent and all applicable local laws. Data sharing should never compromise participant privacy. It is therefore not appropriate to publicly share personally identifiable data on human research participants. The following are examples of data that should not be shared:

-Location data

Response: These are elephants, not human patients and the identifying information are relevant to the viral dynamics within the herd. Approval to include this demographic data was obtained from the veterinary departments responsible for the care of the elephants. If the demographic data is still in breach of the PLoS ONE data policy, I can remove the demographic table, however I believe it holds relevant information to the reader.

Response: One additional study was recommended to cite, and has been included due to its use in drawing comparisons in viral loads and clinical progression (reference [62]). A second additional reference has been added to reference treatment protocols used [58].

Response: No other changes to the reference list were required, nor were any references to papers that have been retracted.

Reviewer 1:

The manuscript by Wheelahan et al describes a lethal hemorrhagic disease case caused by EEHV1B in an elephant that resided in an Australian zoo. In addition to clinical findings, a complete genome sequence of EEHV1B was determined from heart tissue of the deceased animal. Few cases of EEHV1B have been described and correspondingly, there is little information available for the genomic landscape of EEHV1B strains or genotypes. Overall, this is a well written, comprehensive description of a rare case of EEHV1B-associated hemorrhagic disease and addition of another 1B genome sequence is an important contribution. While there is detailed discussion of finer aspects and peculiarities of the genome sequence, much of the speculation is not inherently novel above many of the observations put forth by several publications from the Hayward lab group.

Thank you for taking the time to review this report. We appreciate the fair and succinct review.

Minor points:

1. page 24, line 516 “ age is an unreliable indicator”. This sentence is somewhat misleading and should modified. While the elephant in this case was just out of the normal range of observed EEHV HD cases, the age range 2-8 is still a predominant feature of EEHV HD. In addition, some discussion should be added as to why some elephants could be outliers from this age range.

Response: We concur that age is still a reasonable indicator of risk, in the absence of serology data which provides a more individualised risk profile. The manuscript has been reworded to reflect this. Additionally, we have added further discussion about why elephants may be outliers as listed below:

L562: “… age alone may not reflect individual risk of EEHV-HD”

L570: “This correlation between age and antibodies is likely due to EEHV exposure risk accruing with age and cycles of lytic infection due to reactivation [61]. Disease occurrence in older elephants is likely caused by delayed novel exposure to a species of EEHV, which may occur due to prolonged periods of latency in herd-mates, small herd sizes resulting in less potential for exposure, or older contact with new elephants latently infected with novel EEHVs [13, 14, 26].

2. page 25, lines 552-553, “EEHV1B shedding in trunk wash samples before and after a clinical case of EEHV1B disease has not been described previously…..”

a. EEHV1B trunk wash shedding has been described before the current study in Fuery et al, 2016, JZWM (Clinical Infection of Two Captive Asian Elephants (Elephas Maximus) With Elephant Endotheliotropic Herpesvirus 1b).

b. In fact, the authors should consider comparing observations from that study to the current one in either the intro or discussion or both.

Response: Thank you for the recommended additional reference. We acknowledge that this is a useful study to draw parallels in EEHV1B viral dynamics from and have added additional detail in the discussion to reflect this (New citation [62]). We compare viral dynamics and clinical detail to these cases, as well as exploring possible causes for differences in clinical outcomes (L587-596). We agree that this represents documentation of EEHV1B trunk wash viral dynamics, but importantly the prior study examined trunk wash shedding in only the affected individuals and not across the broader herd following the viraemias (L620-623).

Reviewer 2:

The manuscript provides valuable information on the first reported case of EEHV1B infection in zoo Asian elephants in Australia, with important genetic insights that will interest the readership. However, several points require clarification or expansion to strengthen the manuscript. Please address the following comments to improve clarity, accuracy, and completeness.

Thank you for taking the time to review this report. We appreciate the fair and thorough review.

Title: As the manuscript primarily focuses on genetic characterization, I encourage

revising the title to more clearly reflect the genetic or whole-genome analysis aspect

of this first EEHV1B case in Australia.

Response: Thank you for the suggestion to revise the title. We have now reworded as follows:

“Virological investigation and comparative genomic analysis of elephant endotheliotropic herpesvirus 1B infection in an Australian captive herd of Asian elephants (Elephas maximus)”

General Comments: The manuscript inconsistently uses multiple identifiers for the

same individual/sample (e.g., Elephant 7, AUP_01, EEHV1B_AUP_01_2023).

Please standardize terminology throughout to avoid confusion.

Response: To clarify, Elephant 7 refers to the elephant affected by EEHV-HD, whereas EEHV1B_AUP_01_2023 refers to the EEHV1B genome derived from whole genome sequencing on heart tissue of this elephant, and AUP_01 is an abbreviated identifier for in-text brevity.

The nomenclature used in this paper is as follows: within the abstract and at the first introduction of the genome the entire identifier is used, whereas any in-text reference (including in subheadings, figure captions) uses the abbreviated AUP_01. Figures have been updated to use AUP_01, and for the SplitsTree figures which compare to other EEHV sequences, the figures have been updated t

---

## [Editor Report · Decision Letter 1]

30 Apr 2026

Virological investigation and comparative genomic analysis of elephant endotheliotropic herpesvirus 1B infection in an Australian captive herd of Asian elephants (*Elephas maximus*)

PONE-D-26-06285R1

Dear Dr. Wheelahan,

We’re pleased to inform you that your manuscript has been judged scientifically suitable for publication and will be formally accepted for publication once it meets all outstanding technical requirements.

Kind regards,

Pierre Roques, Ph.D.

Academic Editor

PLOS One

Additional Editor Comments (optional):

Nice work, thank you to have chosen PlosOne for publication
---

## [Editor Report · Acceptance letter]

PONE-D-26-06285R1

PLOS One

Dear Dr. Wheelahan,

I'm pleased to inform you that your manuscript has been deemed suitable for publication in PLOS One. Congratulations! Your manuscript is now being handed over to our production team.

Kind regards,

on behalf of

Dr. Pierre Roques

Academic Editor

PLOS One